# OPTIMIZING WHAT MATTERS: AUC-DRIVEN LEARNING FOR ROBUST NEURAL RETRIEVAL

## ABSTRACT

Dual-encoder retrievers depend on the principle that relevant documents should score higher than irrelevant ones for a given query. Yet the dominant Noise Contrastive Estimation (NCE) objective, which underpins Contrastive Loss, optimizes a softened ranking surrogate that we rigorously prove is fundamentally oblivious to score separation quality and unrelated to AUC. This mismatch leads to poor calibration and suboptimal performance in downstream tasks like retrieval-augmented generation (RAG). To address this fundamental limitation, we introduce the MW loss, a new training objective that maximizes the Mann-Whitney U statistic, which is mathematically equivalent to the Area under the ROC Curve (AUC). MW loss encourages each positive-negative pair to be correctly ranked by minimizing binary cross entropy over score differences. We provide theoretical guarantees that MW loss directly upper-bounds the AoC, better aligning optimization with retrieval goals. We further promote ROC curves and AUC as natural threshold-free diagnostics for evaluating retriever calibration and ranking quality. Empirically, retrievers trained with MW loss consistently outperform contrastive counterparts in AUC and standard retrieval metrics. Our experiments show that MW loss is an empirically superior alternative to Contrastive Loss, yielding better-calibrated and more discriminative retrievers for high-stakes applications like RAG.

## 1 INTRODUCTION

Retrieval-augmented generation (RAG) has rapidly become the standard architecture for knowledge-intensive NLP, powering applications such as web search, enterprise question answering, and data analysis copilots (Lewis et al., 2020; Gao et al., 2023). At the heart of any RAG pipeline is a *dense neural retriever*, which provides the critical initial step of selecting relevant passages based on similarity scores. The reliability and accuracy of these scores directly influence the quality of retrieval outcomes, emphasizing the importance of having well-calibrated retrievers to avoid propagating irrelevant or misleading information.

Dual-encoder models trained with *contrastive objectives*, such as InfoNCE (Oord et al., 2018), dominate current retriever training. Popular dual-encoder architectures like DPR (Karpukhin et al., 2020), GTR (Ni et al., 2021), and E5 (Wang et al., 2022) embed queries and passages in a shared vector space, where retrieval occurs by ranking passages via cosine similarity. However, InfoNCE optimizes only the *relative* ordering of positive and negative examples per query, ignoring global score consistency across queries. Consequently, scores produced by contrastively trained retrievers cannot be meaningfully compared or thresholded globally, limiting their suitability for real-world RAG deployments.

A natural and principled approach for assessing retriever calibration is the *Receiver Operating Characteristic (ROC) curve* and its associated *Area Under the Curve (AUC)*. ROC curves represent the true-positive rate versus the false-positive rate across different score thresholds; a higher AUC (equivalently, lower Area-over-the-Curve, AoC) indicates clearer separation between relevant and irrelevant documents. Crucially, AUC is mathematically identical to the *Mann–Whitney U-statistic* (Mann & Whitney, 1947), measuring the probability that a randomly chosen relevant document scores higher than an irrelevant one. As illustrated in Figure 1, optimizing directly for AUC encourages stronger global separation between positive and negative score distributions.

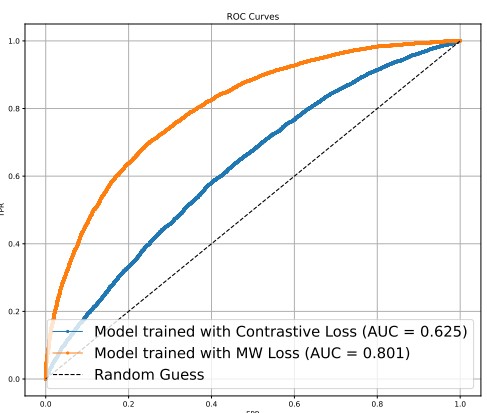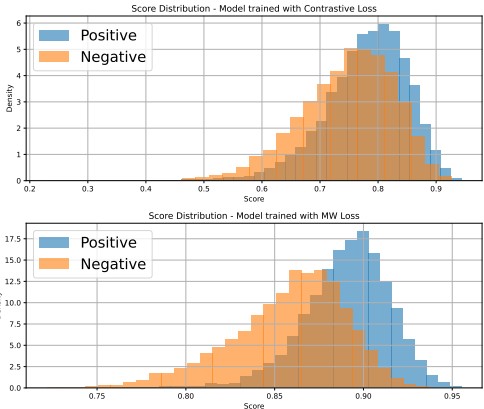

Figure 1: Histogram of positive and negative scores by models trained on NLI dataset using Contrastive loss and MW loss. Model trained with MW loss, creates better separation of scores distribution and its ROC curve dominates the ROC of the model trained with contrastive loss everywhere.

Motivated by these insights, we hypothesize that directly optimizing an AUC-aligned loss function will yield retrievers with better global score calibration, leading to improved retrieval metrics. To address this, we propose **Mann–Whitney (MW) loss**, a novel training objective explicitly designed to maximize the AUC. MW loss minimizes binary cross-entropy over pairwise differences between positive and negative document scores, directly encouraging correct ranking across the entire distribution, not just within-query batches. We provide theoretical guarantees that MW loss upper-bounds the AoC, thus explicitly aligning optimization with the ideal retrieval goal. Empirically, Figure 1 demonstrates how MW loss achieves greater separation between positive and negative scores compared to InfoNCE under identical training conditions.

Our experiments on several open-domain benchmarks consistently show that retrievers trained with MW loss outperform their contrastively trained counterparts in calibration metrics (AUC) and conventional retrieval metrics (MRR, nDCG).

Our primary contributions are as follows:

- We study the problem of global thresholding in dense retrievers and argue the current shortcoming stems from the current training objective used predominently (Karpukhin et al. (2020), Ni et al. (2021), Wang et al. (2022)). In doing so we expose a blind spot of the INfoNCE(Oord et al. (2018)) loss.

- Borrowing from the AUC optimization literature (Gao & Zhou (2012)), we introduce the MW loss, a simple, AUC-aligned objective with provable theoretical bounds on AoC, promoting better global separation between relevant and irrelevant passages.

- We validate MW loss across multiple retrieval benchmarks, demonstrating superior calibration and improved retrieval performance.

## 2 RELATED WORK

**Contrastive representation learning for retrieval.** Noise Contrastive Estimation (NCE) was originally proposed for estimating unnormalised parametric distributions (Ma & Collins, 2018). Its modern variant, InfoNCE, has been rediscovered as a general-purpose representation learner (Oord et al., 2018) and shown to maximise a lower bound on mutual information (Belghazi et al., 2018; Oord et al., 2018). InfoNCE (and contrastive loss more broadly) now underpins most dense retrievers in NLP, including DPR (Karpukhin et al., 2020), GTR (Ni et al., 2021), E5 (Wang et al., 2022), and Contriever (Izacard et al., 2022). Similar ideas have also bridged modalities, as illustrated by CLIP (Radford et al., 2021), demonstrating the versatility of contrastive objectives for cross-modal retrieval. We found, Xiong et al. (2020), which has used metric learning representation learning, somewhat close to our idea as it uses pairwise comparisons. However, the pairwise comparisons are conditioned on the query, which leaves the training oblivious to AUC.

| Step | Contrastive Loss | MW Loss |
|------|------------------|---------|
| Embedding | $2B$ | $2B$ |
| Similarity | $B^2$ | $B^2$ |
| Comparisons | $B(B-1)$ | $B^2(B-1)$ |

(a) Computation Comparison  (b) Contrastive Loss  (c) MW Loss

Figure 2: **Visual Comparison of Contrastive Loss vs. MW Loss**. The MW Loss performs more pairwise comparisons without increasing the embedding or similarity computation cost. In Figures 2b and 2c, each square in the colored matrix represents a similarity computation between query and passage (green for a positive pair, red for a negative pair). Similarity scores are aggregated differently for each loss function, converging to a grey square above. Each grey square is then summed up to obtain the final loss.

**Enhancing retriever training dynamics.** Multiple studies refine how contrastive retrievers are trained. Momentum encoders such as MoCo (He et al., 2020) and BYOL (Grill et al., 2020) update a slow-moving target network to stabilise learning. The choice and curation of negative examples further influence convergence: hard-negative mining (Cai et al., 2022; Gao et al., 2021; Moreira et al., 2024) makes the task more discriminative and accelerates optimisation. Beyond data curriculum, several works modify the loss itself, either framing each pair as a binary classification (Zhai et al., 2023; Zhuang et al., 2023), adding separation-promoting regularisers (Moiseev et al., 2023; Wang et al., 2024), or combining both signals(Zhang et al., 2024).

**AUC-aligned objectives and rank statistics.** Our work is inspired by classical rank statistics such as the Mann–Whitney U-test (Mann & Whitney, 1947; Fawcett, 2006) and by metric-learning research that explicitly targets margin-based separation (Kulis et al., 2013). Optimising triplet loss (Hoffer & Ailon, 2015; Burges et al., 2005; Zhuang et al., 2023) and, more recently, directly maximising AUROC (Zhu et al., 2022) are examples of aligning training objectives with AUC evaluation criteria. Our theretrical guarantees are directly inspired by the work of Gao & Zhou (2012), which comprehensively studies the loss functions and conditions for their consistency with AUC. In our work we start from contrastive loss and modify it so it becomes consistent with AUC, the way it is measured for retrieval problems.

## 3 PROBLEM STATEMENT

### 3.1 BACKGROUND

Let $\{s_1^+, \ldots, s_{n_+}^+\}$ denote the scores a retriever assigns to relevant passages and $\{s_1^-, \ldots, s_{n_-}^-\}$ the scores for irrelevant passages. The *Wilcoxon–Mann–Whitney* (MW $U$) rank–sum statistic (Mann & Whitney, 1947) counts the number of positive–negative pairs that are correctly ordered:

$$U = \sum_{i=1}^{n_+} \sum_{j=1}^{n_-} \mathbf{1}(s_i^+ > s_j^-)$$

where $\mathbf{1}(\cdot)$ is the indicator function. Normalising $U$ by the total number of pairs yields the *area under the ROC curve* (AUC):

$$\text{AUC} = \frac{U}{n_+ n_-} = \Pr(s^+ > s^-)$$

Thus AUC is the probability that a randomly chosen positive sample scores higher than a randomly chosen negative sample (Fawcett, 2006). This probabilistic interpretation of the AUC paves the way for theoretical study of learning algorithms and their relation to AUC (Gao & Zhou (2012)).

| Query | Passage (excerpt) | Score |
|---|---|---|
| Why is the sky blue? | **The sky appears blue because of a phenomenon called Rayleigh scattering...** | 0.85 |
| | At sunrise and sunset, the sky often turns vibrant shades of red, orange, and pink... | 0.83 |
| | The Earth's atmosphere is composed mainly of nitrogen and oxygen... | 0.83 |
| Why are seasons different between Earth semispheres? | **Throughout the year, the Sun's rays strike different parts of Earth more directly...** | 0.82 |
| | Earth takes about 365.25 days to complete one orbit around the Sun... | 0.80 |
| | The equator receives more consistent sunlight throughout the year... | 0.79 |

Figure 3: Examples where irrelevant passages receive similar scores to relevant ones, making threshold-based filtering unreliable. Relevant passages are in bold.

## 3.2 LIMITATIONS OF THE CONTRASTIVE LOSS

Dense retrievers are trained with the Contrastive Loss equation 1 to rank the documents based on their relevance for a given query. This is done through learning a metric or score function which assigns a similarity score to each pair. This is achieved through Learning an encoder function which separately encodes the query and document and applying a computationally cheap function on top of these two embeddings. This separation is necessary to allow offline indexing of the documents. This technique was first introduced in Karpukhin et al. (2020). In Karpukhin et al. (2020), this view of the retriever problem was thought of as a metric learning problem (Kulis et al., 2013), and proposed to use a tailord version of the Noise Contrastive Loss equation 1 as a learn to rank objective.

Contrastive Loss has limitations for learning a general metric function. This can be observed by the fact that the loss function is invariant under shifting scores with a constant value for a given query. This phenomenon is illustrated in Figure 3. The first query's irrelevant passages have a score higher than the relevant passage of the second query. This problem could have been mitigated by shifting all the scores of the second query 0.03 points.

We will now more formally discuss how AUC is a blind spot for Contrastive Loss. Consider the general form of Contrastive Loss for training dual encoders as below:

Where the $\mathcal{Q}$, $\mathcal{P}^+(\cdot \mid q)$ and $\mathcal{P}^-(\cdot \mid q)$ denote the distributions of queries, distribution of positive and negative passages conditioned on a query respectively.

$$\mathcal{L}_{\text{CL}} = -\mathbb{E}_{q\sim\mathcal{Q},\, p^+\sim\mathcal{P}^+(\cdot|q),\, \{p_k^-\}_{k=1}^K \overset{\text{i.i.d.}}{\sim} \mathcal{P}^-(\cdot|q)} \left[ \log \frac{\exp(\text{sim}(q,p^+)/\tau)}{\exp(\text{sim}(q,p^+)/\tau) + \sum_{k=1}^K \exp(\text{sim}(q,p_k^-)/\tau)} \right] \tag{1}$$

**Lemma 1** (Shift-invariance & unconstrained AoC for Contrastive Loss). *Let's define:*

$$\ell_\tau(s^+, S^-) = -\log\frac{e^{s^+/\tau}}{e^{s^+/\tau} + \sum_{s^-\in S^-} e^{s/\tau}}, \qquad \tau > 0.$$

*Where $s^+$ is a positive score and $S^-$ is a set of negative scores. With notations of equation equation 1, we define $s^+ = s(q,p^+)$ and $S^- = \{s(q,p^-) \mid p^- \in \{p_k^-\}_{k=1}^K\}$. Placing these into definition of $\ell_\tau$, the population loss can be rewritten as:*

$$\mathcal{L}_\tau[s] = \mathbb{E}_{q\sim\mathcal{Q},\, p^+\sim\mathcal{P}^+(\cdot|q), \{p_k^-\}_{k=1}^K \overset{\text{i.i.d.}}{\sim} \mathcal{P}^-(\cdot|q)} \left[ \ell_\tau(s^+, S^-) \right].$$

1. ***Shift-invariance.*** *For any measurable offset $g\colon \mathcal{Q}\to\mathbb{R}$, define the shifted scorer*

$$s_g(q,d) = s(q,d) + g(q).$$

*Then $\mathcal{L}_\tau[s_g] = \mathcal{L}_\tau[s]$.*

2. ***Arbitrary degradation of AoC.*** *If $|s(q,d)| \leq M < \infty$, then for every $\varepsilon > 0$ there exists an offset $g$ such that the Area-over-ROC (AoC) defined as :*

$$\text{AoC}[s] = \Pr_{q_1,q_1\sim\mathcal{Q},p^+\sim\mathcal{P}^+(.|q_1),p-\sim\mathcal{P}^-(.|q_2)} \left[ s(q_1,p^+) < s(q_2,p^-) \right]$$

*satisfies* $\text{AoC}[s_g] \geq 0.5 - \varepsilon$. *Hence global positive–negative separation can be made arbitrarily poor without altering the Contrastive Loss.*

We are now set to propose a loss function which is not oblivious to the AUC metric. We hypothesize that this kind of loss would produce a metric function that in addition to better score separation, performs better on the retrieval metrics.

## 4  PROPOSED METHOD

In order to train a retriever which learns an absolute metric that is not relative to query, we propose a loss function which essentially compares positive and negative pairs one by one. Theoretically this loss provides an upper bound for AoC and we call it **MW** (**M**ann-**W**hitney) loss. This loss has similarity with Contrastive Loss, in that it also makes comparison between positive and negative pairs and encourages the positive score to stand above the negative score. Since in this objective we are comparing every positive pair with every negative pair in a one-on-one fashion, the Cross Entropy formulation of the Contrastive Loss changes to binary classification. Using the same notation as before, MW can be expressed as below:

$$\mathcal{L}_{\text{MW}} = \mathbb{E}_{q_1,q_2 \sim \mathcal{Q}, p^+ \sim \mathcal{P}^+(.|q_1), p^- \sim \mathcal{P}^-(\cdot \,|\, q_2)} \left[ -\log \sigma\big(s(q_1, p^+) - s(q, p^-)\big) \right] \qquad (2)$$

Note that in the definition above, two random (potentially distinct) queries are sampled $(q_1, q_2)$.

The following lemma shows how minimizing **MW** minimizes the area over the curve and hence remedies the blind spot of the Contrastive Loss. This result borrows from the AUC optimizaion literature (Gao & Zhou (2012)), we provide a simpler minimal proof tailored to our specific setup.

**Lemma 2** (MW upper–bounds AoC)**.** *Let D be the data distribution over independent positive and negative instances. Also let us define $\ell_{BCE}$ as:*

$$\ell_{\text{BCE}}(z) = \log\big(1 + e^{-z/\tau}\big), \ \tau > 0.$$

*For a scoring function $s \colon \mathcal{Q} \times \mathcal{P} \to \mathbb{R}$ and using the same notation as before for the distribution of queries, positive passages and per query and negative passages per query, define* MW *loss as :*

$$\mathcal{L}_{\text{MW}}[s] \ = \ \mathbb{E}_{q_1,q_2 \sim \mathcal{Q}, p^+ \sim \mathcal{P}^+(.|q_1), p^- \sim \mathcal{P}^-(\cdot \,|\, q_2)} \big[ \ell_{\text{BCE}}\big(s(q_1, p^+) - s(q_2,{}^-)\big) \big],$$

*Also let us define AoC as in Lemma 1 [1]:*

$$\text{AoC}[s] \ = \ \Pr_{q_1,q_2 \sim \mathcal{Q}, p^+ \sim \mathcal{P}^+(.|q_1), p- \sim \mathcal{P}^-(.|q_2)} \big[ s(q_1, p^+) < s(q_2, p^-) \big]$$

*Then for every scoring function s:*

$$\text{AoC}[s] \ \leq \ \frac{1}{\log 2} \, \mathcal{L}_{\text{MW}}[s]$$

*Proof.* For any real $z$ we have the point-wise inequality

$$\mathbf{1}\{z \leq 0\} \ \leq \ \frac{\log\big(1 + e^{-z/\tau}\big)}{\log 2} \ = \ \frac{\ell_{\text{BCE}}(z)}{\log 2} \qquad (*)$$

Indeed, if $z > 0$ both sides are 0; otherwise $z \leq 0$ and $\ell_{\text{BCE}}(z) \geq \log 2$.

Taking expectations of $(*)$ with $z = s(q_1, p^+) - s(q_2, P^-)$ gives:

$$\text{AoC}[s] \ = \ \mathbb{E}\big[\mathbf{1}\{z \leq 0\}\big] \ \leq \ \frac{1}{\log 2} \, \mathbb{E}\big[\ell_{\text{BCE}}(z)\big] \ = \ \frac{\mathcal{L}_{\text{MW}}[s]}{\log 2}$$

$\square$

---

[1] Note that this is the probability of the event that $s(q_1, p^+) < s(q_2, p^-)$ which is equivalent to the expectation of the random variable $I = \mathbf{1}\{s(q_1, p^+) < s(q_2, p^-)\}$. This alternative interpretation is used in the proof of this lemma.

Thus minimizing the MW loss *directly maximizes* AUC.

Now we shall introduce some notation to express how this loss is approximated within one batch of training. We consider the general setting of training retrievers with a batch size $B$, where each batch contains a set of $B$ queries $\{q_1, q_2, ..., q_b\}$ and one relevant (positive) passage per query $\{p_1^+, p_2^+, ..., p_B^+\}$ and $H$ additional hard negative passages $\{p_{i,1}^-, p_{i,2}^-, ...p_{i,H}^-\}$ per query where $1 \leq i \leq B$. Let $s_i^+ = sim(E(q_i), E(p_i^+))$ to denote the similarity score between the query ($q_i$) and it's corresponding positive document. Also let us by $S_i^-$ denote the set which has the similarity score between the $q_i$ and all passages in the batch except for the relevant passage ($p_i^+$). The set $S_i^-$ would be of size $HB + (B - 1)$ which is comprised of $HB$ elements for all the hard negatives and $B - 1$ for the positive passages of the other queries. Finally, we use the set $S^- = \bigcup_{i=1}^{B} S_i^-$ to denote the set of all negative scores in one batch of training.

With this notation, our proposed loss function would look as below:

$$\mathcal{L}_{\text{MW}} = -\frac{1}{B} \sum_{i=1}^{B} \sum_{s_k^- \in S^-} \log \sigma(s_i^+ - s_k^-) \tag{3}$$

For comparison we add the Contrastive Loss below using the same notation:

$$\mathcal{L}_{CL} = -\frac{1}{B} \sum_{i=1}^{B} \log \left( \frac{\exp(s_i^+/\tau)}{\exp(s_i^+/\tau) + \sum\limits_{s \in S_i^-} \exp(s/\tau)} \right) \tag{4}$$

Figure 2 visualizes the compuational differences between the Contrastive Loss and the MW Loss in a scenario without hard negatives. $B^2$ similarity scores need to be calculated for both methods. MW Loss compares every positive pair with all non-diagonal elements ($B(B-1)$) whereas contrastive loss compares it only with negative scores within the same row ($B$).

## 5 EXPERIMENTS

In this section, we experimentally evaluate our novel Mann-Whitney Loss (MW-Loss), inspired by the Mann-Whitney U statistic, against the conventional Contrastive Loss (CL). We systematically assess both in-distribution performance and out-of-distribution generalization across diverse datasets. To the best of our knowledge, this is the first work to propose training dense neural retrievers in natural language to learn a global metric that is not conditioned on the query. All previous methods differ in strategies such as backbone architecture, training data, and negative sampling, making direct head-to-head comparisons difficult and potentially misleading. We therefore take CL, the predominant objective underlying these methods, as a representative baseline and focus our experiments on comparing MW directly against CL.

### 5.1 EXPERIMENTAL SETUP

We initialize all experiments with pre-trained foundational models to isolate the specific effects of our proposed loss function, eliminating confounding variables. We employ three model archi-tectures across three different sizes, small base and large: MiniLM, XLM-RoBERTa-Base and XLM-RoBERTa-Large (Wang et al., 2020; Devlin et al., 2019; Conneau et al., 2019). For training, we utilize four distinct datasets: Natural Questions (NQ) (Kwiatkowski et al., 2019), Natural Language Inference (NLI) (Gao et al., 2021), SQUAD Rajpurkar et al. (2016), and MS MARCO (Bajaj et al., 2016), each representing different language understanding tasks.

We train each model for a maximum of 20 epochs, with early stopping when the evaluation loss plateaus. We train using a temperature of 0.01 for both Contrastive loss and MW loss functions across all experiments using 5 hard negatives. For all evaluations, we track three key metrics: Area Under the ROC Curve (AUC), Mean Reciprocal Rank at 10 (MRR@10) (Voorhees et al., 1999), and normalized Discounted Cumulative Gain at 10 (nDCG@10) (Järvelin & Kekäläinen, 2002). All

Table 1: Retrieval performance of models trained with contrastive (CL) and Mann-Whitney (MW) loss across datasets. For each dataset, we report AUC, MRR, and nDCG metrics. Average column shows mean across datasets.

| Data | NLI | | | NQ | | | SQuAD | | | MSMarco | | | Average | | |
|------|-----|-----|------|-----|-----|------|-------|-----|------|---------|-----|------|---------|-----|------|
| Loss | AUC | MRR | nDCG | AUC | MRR | nDCG | AUC | MRR | nDCG | AUC | MRR | nDCG | AUC | MRR | nDCG |
| **MINILM** | | | | | | | | | | | | | | | |
| CL | 0.67 | 0.24 | 0.29 | 0.70 | 0.35 | 0.46 | 0.72 | **0.39** | **0.48** | 0.67 | 0.36 | **0.44** | 0.85 | 0.52 | 0.58 |
| MW | **0.81** | **0.35** | **0.43** | **0.84** | **0.36** | 0.43 | **0.85** | 0.37 | 0.44 | **0.81** | 0.34 | 0.43 | **0.91** | **0.53** | 0.58 |
| **XLM-RoBERTa-Base** | | | | | | | | | | | | | | | |
| CL | 0.69 | 0.27 | 0.32 | 0.70 | 0.36 | 0.44 | 0.78 | 0.38 | 0.47 | 0.69 | 0.27 | 0.32 | 0.84 | 0.53 | 0.6 |
| MW | **0.84** | **0.37** | **0.45** | **0.87** | 0.36 | **0.45** | **0.86** | 0.36 | 0.45 | **0.84** | 0.37 | **0.45** | **0.93** | **0.55** | **0.61** |
| **XLM-RoBERTa-Large** | | | | | | | | | | | | | | | |
| CL | 0.73 | 0.31 | 0.37 | 0.79 | 0.38 | 0.48 | 0.65 | 0.31 | 0.38 | 0.73 | 0.31 | 0.37 | 0.84 | 0.54 | 0.58 |
| MW | **0.88** | 0.39 | **0.47** | **0.89** | 0.37 | 0.46 | **0.92** | **0.41** | **0.50** | **0.87** | **0.39** | **0.47** | **0.96** | **0.59** | **0.64** |

experiments were carried out on two NVIDIA A100 GPU with batch sizes of 16, 32 and 64 for large, base and small model variants, respectively. In all our trainings we used 500 steps of warmup.

For the computation of AUC, we employ a comprehensive evaluation protocol: for each query, we consider the scores of positive passages as positive samples. To obtain reliable negative samples, we compute the similarity score between the query and all passages in the corpus, selecting the top 500 highest-scoring passages (excluding known positives) as hard negatives. The positive and negative scores from all queries are then aggregated into a single pool, from which we calculate the final AUC metric. This approach ensures that our AUC evaluation captures model performance across the most challenging cases in the entire dataset.

## 5.2 IN-DISTRIBUTION PERFORMANCE

First, we evaluate models trained and tested on the same dataset to establish baseline performance. For each dataset-model combination, we report the performance across our tracked metrics to quantify the effectiveness of both loss functions in the standard setting.

Based on Table 1, the comparison between models trained with contrastive loss (CL) and MW loss reveals several key insights. The MiniLM and Roberta-base models using MW loss perform on par (on average) with CL across retrieval metrics (MRR and nDCG), but notably improves upon AUC scores across all datasets. For the RoBERTa-Large model, we observe improvements across all metrics when using the MW loss. These results suggest that Mann-Whitney loss provides a consistent advantage for ranking performance, with in-domain benefits becoming more pronounced as model size increases, having a higher capacity for learning the harder objective.

## 5.3 CROSS-DATASET GENERALIZATION

To rigorously evaluate generalization capabilities, we trained models on the Natural Language Inference (NLI) dataset—selected for its extensive size and semantic diversity—and assessed their transfer performance across the challenging BEIR benchmark suite Thakur et al. (2021). This comprehensive evaluation protocol enabled systematic analysis of how effectively MW-Loss and contrastive learning (CL) loss facilitate knowledge transfer to unseen domains and tasks.

As shown in Table 2, MW-Loss consistently outperforms conventional CL loss across multiple retrieval metrics. These empirical results align with our theoretical guarantees, confirming that directly optimizing the AUC by minimizing MW loss translates to practical advantages in both zero-shot and in-domain generalization scenarios.

Figure 4 illustrates the performance gains for each evaluation metric across datasets. Interestingly, we observe that these improvements remain consistent regardless of model size and type, suggesting that MW-Loss provides representational benefits independent of parameter count.

Table 2: **Comparison of CL and MW losses across unseen BEIR retrieval and SQUAD datasets**. Performance is measured using AUC, MRR, and nDCG metrics for models of different sizes and types trained on the NLI dataset. The results show that MW outperforms CL in the majority of datasets and metrics.

| Dataset | Loss | MiniLM | | | XLM-RoBERTa-Base | | | XLM-RoBERTa-Large | | |
|---|---|---|---|---|---|---|---|---|---|---|
| | | AUC | MRR | nDCG | AUC | MRR | nDCG | AUC | MRR | nDCG |
| **NFCorpus** | CL | 0.49 | 0.26 | 0.16 | 0.51 | 0.27 | 0.18 | 0.56 | 0.28 | 0.19 |
| | MW | **0.50** | 0.26 | **0.17** | **0.55** | **0.30** | **0.20** | **0.58** | 0.31 | 0.20 |
| **Trec-Covid** | CL | **0.20** | 0.34 | 0.19 | 0.24 | 0.37 | 0.25 | 0.29 | 0.39 | 0.24 |
| | MW | 0.16 | **0.44** | **0.22** | **0.26** | **0.43** | **0.31** | **0.28** | **0.43** | **0.38** |
| **FiQA** | CL | 0.44 | 0.12 | 0.09 | 0.43 | 0.14 | 0.10 | **0.52** | 0.18 | 0.12 |
| | MW | **0.48** | **0.16** | **0.11** | **0.49** | **0.17** | **0.12** | 0.51 | 0.18 | **0.13** |
| **SQUAD** | CL | 0.80 | 0.40 | 0.44 | 0.85 | 0.47 | 0.51 | 0.86 | 0.47 | 0.52 |
| | MW | **0.86** | **0.41** | **0.45** | **0.90** | **0.52** | **0.90** | **0.92** | **0.57** | **0.62** |
| **Hotpot QA** | CL | 0.66 | 0.62 | 0.46 | 0.64 | 0.55 | 0.41 | 0.66 | **0.63** | **0.48** |
| | MW | **0.69** | 0.62 | **0.47** | **0.70** | **0.56** | **0.46** | **0.72** | 0.59 | 0.46 |
| **Touche-2020** | CL | **0.27** | **0.12** | **0.07** | 0.24 | 0.08 | 0.05 | 0.37 | 0.12 | 0.05 |
| | MW | 0.25 | 0.09 | 0.06 | **0.26** | **0.09** | **0.07** | **0.39** | **0.18** | **0.07** |
| **ArguAna** | CL | 0.71 | 0.15 | 0.21 | 0.83 | **0.17** | 0.24 | 0.86 | **0.19** | 0.27 |
| | MW | **0.77** | **0.16** | **0.22** | **0.86** | 0.15 | **0.27** | **0.90** | 0.17 | **0.31** |
| **CQADupStack** | CL | 0.38 | 0.19 | 0.17 | 0.26 | 0.13 | 0.11 | 0.45 | 0.28 | 0.25 |
| | MW | **0.40** | **0.20** | **0.18** | **0.33** | **0.19** | **0.16** | **0.52** | **0.33** | **0.31** |
| **Quora** | CL | 0.94 | 0.86 | 0.85 | 0.94 | 0.83 | 0.81 | 0.97 | 0.87 | 0.86 |
| | MW | **0.97** | 0.86 | 0.85 | **0.96** | **0.86** | **0.84** | **0.98** | **0.88** | 0.86 |
| **DBPedia** | CL | 0.40 | 0.36 | 0.24 | 0.32 | 0.23 | 0.14 | 0.56 | 0.33 | 0.23 |
| | MW | **0.42** | 0.36 | 0.24 | **0.46** | **0.30** | **0.20** | **0.59** | **0.38** | **0.29** |
| **Scidocs** | CL | 0.31 | 0.10 | 0.05 | 0.29 | 0.12 | 0.06 | 0.42 | 0.17 | 0.07 |
| | MW | **0.38** | **0.12** | **0.06** | **0.30** | 0.12 | 0.06 | **0.43** | **0.18** | **0.08** |
| **Scifact** | CL | 0.61 | 0.20 | 0.22 | 0.68 | 0.26 | 0.28 | 0.66 | 0.36 | 0.39 |
| | MW | **0.63** | **0.21** | **0.23** | **0.70** | **0.28** | **0.29** | **0.77** | **0.39** | **0.41** |
| **Fever** | CL | 0.47 | 0.18 | 0.19 | 0.64 | 0.32 | 0.35 | 0.73 | 0.50 | 0.52 |
| | MW | **0.54** | **0.24** | **0.25** | **0.73** | **0.44** | **0.45** | **0.75** | **0.53** | **0.54** |
| **Climate-Fever** | CL | 0.50 | 0.16 | 0.12 | **0.62** | **0.32** | **0.23** | 0.67 | 0.38 | 0.29 |
| | MW | **0.53** | **0.19** | **0.13** | 0.58 | 0.27 | 0.20 | **0.70** | **0.40** | **0.32** |
| **Average** | CL | 0.51 | 0.29 | 0.25 | 0.54 | 0.30 | 0.27 | 0.61 | 0.37 | 0.32 |
| | MW | **0.54** | **0.31** | **0.26** | **0.58** | **0.33** | **0.32** | **0.65** | **0.39** | **0.36** |

# 6  CONCLUSION

In this work, we revisited the foundational objective of dense retriever training and exposed a critical shortcoming in the widely used Contrastive Loss: its inability to promote globally calibrated scores, due to its invariance to query-specific score shifts. This misalignment hinders the application of contrastively trained retrievers in real-world scenarios that demand consistent and thresholdable relevance scores. To remedy this, we proposed the Mann–Whitney (MW) loss, a simple yet principled objective that directly maximizes the Area Under the ROC Curve (AUC) by minimizing binary cross-entropy over pairwise score differences. We provided theoretical guarantees that MW loss upper-bounds the Area-over-the-Curve (AoC), addressing a key blind spot in contrastive learning.

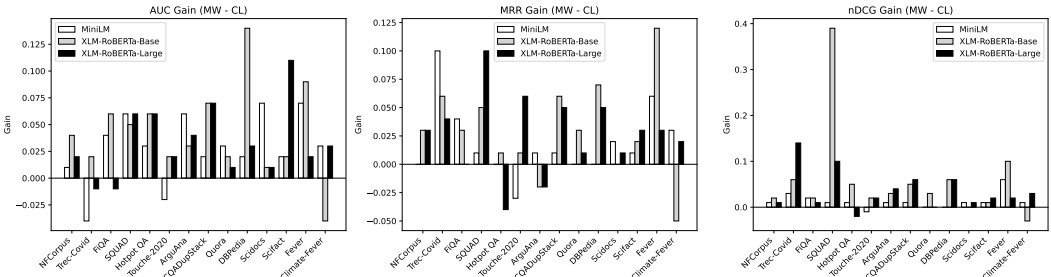

Figure 4: The gain in performance of using MW over CL for across metrics and models. The plots show AUC gain (left), MRR gain (center), and nDCG gain (right). Positive values indicate superior performance of MW compared to CL.

Empirical evaluations across both in-distribution and out-of-distribution benchmarks confirm the effectiveness of MW loss: models trained with MW consistently outperform their contrastive counterparts not only in AUC but also in standard retrieval metrics (on average) such as MRR and nDCG. Notably, these gains persist across model sizes and evaluation domains, highlighting MW loss as a broadly applicable and easily integrable alternative for training robust neural retrievers. We hypothesize that these improvements come as a result of targeting a harder objective, learning a global metric, which opens up the path for model to learn a more generalizable solution, this hypothesize is partially confirmed by the fact that MW suffers from a slower convergence compared to CL (see appendix C). We hope this work motivates a rethinking of retrieval objectives and encourages further exploration into calibration-aware learning for dense retrieval systems.

## 7 LIMITATIONS

While MW loss is a principled and empirically strong alternative to contrastive objectives for dense retriever training, several limitations and open questions remain.

First, although MW loss is defined as a population-level objective, it is approximated within batches. This raises questions about how best to approximate the population loss within limited batch data and whether emphasizing hard negatives improves retrieval performance, both of which may impact training dynamics and generalization.

Second, a gradient level analysis of the MW loss and impact hard negatives could provide theoretical insight into its empirical advantages.

Third, while prior work has studied the geometry of representations under contrastive training (Gao et al., 2019), analyzing MW loss's impact on representation geometry is an open direction.

Lastly, although we show strong results even with modest computational budgets and model sizes, state-of-the-art retrievers like E5 (Wang et al., 2022) are trained using extensive resources across diverse domains. Exploring the full potential of MW loss in such large-scale training regimes—including domain adaptation, multilingual settings, and production deployment—is a promising direction for future work.

## 8 ETHICS STATEMENT

Our objective is to improve calibration and ranking reliability of neural retrievers, which can reduce downstream propagation of irrelevant or misleading content in RAG systems. We explicitly analyze failure modes (score shift invariance) and propose an AUC-aligned objective intended to improve thresholdability and safety of retrieval decisions.

All methods, hyperparameters, training/evaluation protocols, and computing assumptions are documented in the paper and supplementary material. We do not fabricate, falsify, or omit results; negative and neutral outcomes (e.g., slower convergence of MW) are reported. Reproducibility artifacts (code, configs, scripts) will be released upon publication to enable verification.

We use only publicly available datasets under their licenses and do not attempt re-identification or linkage. No new personal data were collected. No confidential, proprietary, or embargoed information was used. All third-party code and datasets are cited.

Benchmarks used (NQ, SQuAD, MS MARCO, BEIR subsets) may encode societal skews. Our contribution focuses on the loss function; it does not itself remove dataset or retrieval bias. We disclose this limitation and recommend auditing retrieved content with group-aware evaluation when sensitive attributes are present. We welcome community feedback and commit to documenting any discovered disparities.

We disclose that there are no conflicts of interest that would bias this work. Any funding sources will be listed in the camera-ready version per ICLR policy.

We disclose that we have used LLMs for the sole purpose of polishing and correcting typos and grammatical errors in this paper.

## 9 REPRODUCIBILITY STATEMENT

We follow ICLR's reproducibility expectations. We will release: (a) source code; (b) exact training/evaluation configs; (c) scripts for data preparation, retrieval, and metrics; (d) fixed seeds and environment specs; (e) model checkpoints at the reported early-stopping iterations.

We provide scripts to fetch and preprocess: NLI, NQ, SQuAD, MS MARCO, and BEIR tasks evaluated in the paper.

For each backbone (MiniLM, XLM-RoBERTa-Base/Large) and loss (CL vs. MW), we include: Optimizer, LR schedule, batch size, temperature(s), warmup steps.

We release evaluation scripts that reproduce all tables/figures:

We provide scripts/configs for ablations over batch size, number of hard negatives, and learning rate. We also include: (i) training-steps-to-best-checkpoint curves; (ii) sensitivity to temperature(s); and (iii) a fairness note encouraging group-aware analyses when datasets contain sensitive attributes.

We release all artifacts including, trained checkpoints for all main results, the exact validation logs used for early stopping and figure-generation notebooks for ROC curves, score histograms, and metric-gain plots.

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

# A  PROOF OF LEMMA 1

**Lemma** (Lemma 1). *Let*

$$\ell_\tau(s^+, S^-) \;=\; -\log \frac{e^{s^+/\tau}}{e^{s^+/\tau} + \sum_{s \in S^-} e^{s/\tau}}, \qquad \tau > 0.$$

*With notations of equation equation 1 the population loss can be rewritten as:*

$$\mathcal{L}_\tau[s] \;=\; \mathbb{E}_{q \sim \mathcal{Q},\, p^+ \sim \mathcal{P}^+(\cdot|q),\, \{p_k^-\}_{k=1}^K \overset{i.i.d.}{\sim} \mathcal{P}^-(\cdot|q)} \Big[ \ell_\tau\big(s(q,p^+), \{s(q,p^-) \mid p^- \in \{p_k^-\}_{k=1}^K\}\big)\Big].$$

1. **Shift-invariance.** *For any measurable offset $g\colon \mathcal{Q} \to \mathbb{R}$, define the shifted scorer*

$$s_g(q,d) \;=\; s(q,d) + g(q).$$

*Then $\mathcal{L}_\tau[s_g] = \mathcal{L}_\tau[s]$.*

2. **Arbitrary degradation of AoC.** *If $|s(q,d)| \leq M < \infty$, then for every $\varepsilon > 0$ there exists an offset $g$ such that the Area-over-ROC (AoC) defined as :*

$$\mathrm{AoC}[s] \;=\; \Pr_{q_1,q_1 \sim \mathcal{Q},\, p^+ \sim \mathcal{P}^+(.|q_1),\, p- \sim \mathcal{P}^-(.|q_2)} \big[s(q_1,p^+) < s(q_2,p^-)\big]$$

*satisfies $\mathrm{AoC}[s_g] \geq 0.5 - \varepsilon$. Hence global positive–negative separation can be made arbitrarily poor without altering the Contrastive Loss.*

*Proof.*

**(i) Shift-invariance.** Fix $q$ and abbreviate $Z = e^{s^+/\tau} + \sum_{s \in S^-} e^{s/\tau}$. Adding $g(q)$ to *every* score gives

$$Z_g = e^{(s^+ + g)/\tau} + \sum_{s \in S^-} e^{(s+g)/\tau} = e^{g/\tau} Z,$$

while the numerator of $\ell_\tau$ is also multiplied by $e^{g/\tau}$. Hence $\ell_\tau$ is unchanged and so is the expectation $\mathcal{L}_\tau$.

**(ii) Arbitrary AoC degradation.** Draw independent offsets $g(q) \sim \mathcal{N}(0, \sigma^2)$. For two independent samples $(q_1, p^+)$ and $(q_2, p^-)$, we have:

$$s_g(q_1, p^+) - s_g(q_2, p^-) = \underbrace{s(q_1, p^+) - s(q_2, p^-)}_{\text{bounded by } 2M} + \underbrace{g(q_1) - g(q_2)}_{\mathcal{N}(0, 2\sigma^2)}.$$

As $\sigma \to \infty$ the Gaussian term dominates, so $\Pr[s_g(q_1, p^+) < s_g(q_2, p^-)] \to 0.5$. Choose $\sigma$ large enough that the probability exceeds $0.5 - \varepsilon$; this yields the required $g$.

$\square$

## B   ABLATION OVER HARD NEGATIVES, LR, AND BATCH SIZE

We conduct controlled ablations on the sensitivity of training on three hyper parameters, namely the learning rate, batch size and number of hard negatives. Our study is limited to training on the MiniLM case. For brevity, we show the results with the best learning rate for each batch size.

| lr | batch_size | hard_negative | precision@10 | recall@1 | MRR | nDCG@10 | AUC |
|---|---|---|---|---|---|---|---|
| 3e-5 | 64 | 3 | 0.10 | 0.36 | 0.55 | 0.63 | 0.90 |
| 3e-5 | 64 | 5 | 0.10 | 0.37 | 0.55 | 0.64 | 0.90 |
| 3e-5 | 64 | 7 | 0.10 | 0.37 | 0.55 | 0.63 | 0.90 |
| 5e-5 | 128 | 3 | 0.10 | 0.36 | 0.55 | 0.63 | 0.89 |
| 5e-5 | 128 | 5 | 0.10 | 0.36 | 0.54 | 0.62 | 0.88 |
| 5e-5 | 128 | 7 | 0.10 | 0.38 | 0.56 | 0.64 | 0.92 |

Table 3: Results across hyperparameter settings for CL loss.

| lr | batch_size | hard_negative | precision@10 | recall@1 | MRR | nDCG@10 | AUC |
|---|---|---|---|---|---|---|---|
| 3e-5 | 64 | 3 | 0.10 | 0.35 | 0.54 | 0.62 | 0.96 |
| 3e-5 | 64 | 5 | 0.10 | 0.35 | 0.54 | 0.63 | 0.96 |
| 3e-5 | 64 | 7 | 0.10 | 0.35 | 0.54 | 0.62 | 0.96 |
| 3e-5 | 128 | 3 | 0.10 | 0.34 | 0.53 | 0.62 | 0.97 |
| 3e-5 | 128 | 5 | 0.10 | 0.34 | 0.53 | 0.62 | 0.96 |
| 3e-5 | 128 | 7 | 0.10 | 0.34 | 0.53 | 0.62 | 0.97 |

Table 4: Results across hyperparameter settings for MW loss.

The tables indicate that MW loss is rather stable with respect to changes in batch size and number of hard negative examples while CL loss is more sensitive.

## C   CONVERGENCE ANALYSIS

In this section we provide results on the number of training steps until the the best checkpoint.

Based on these results MW has a slower convergence rate. While this can be seen as a shortcumming of MW we hypothesize that this is what causes this loss function to perform better. Particularly, we believe that by targeting a harder objective, learning a global metric rather than a conditional, MW targets a stronger objective which is harder to achieve which results in a slower convergence rate, at the same time this stronger objective has better generalization and outperforms the contrastive loss.

Table 5: **Training steps at best checkpoint** for each dataset and loss (MW vs CL), across model sizes.

| Model | NLI | MS MARCO | SQuAD | NQ |
|---|---|---|---|---|
| Small (MW) | 16000 | 5000 | 2000 | 9500 |
| Small (CL) | 14000 | 3500 | 1000 | 7000 |
| Base (MW) | 24000 | 3500 | 4000 | 12500 |
| Base (CL) | 20000 | 1500 | 2500 | 11000 |
| Large (MW) | 21500 | 4500 | 2500 | 10000 |
| Large (CL) | 18000 | 3000 | 2000 | 8000 |

