# OpenReview forum: "Optimizing What Matters: AUC-Driven Learning for Robust Neural Retrieval"
_ICLR.cc/2026/Conference — Submitted to ICLR 2026_

### Official Review · Reviewer_DCGo · 2025-10-18

**Soundness:** 3
**Presentation:** 2
**Contribution:** 3
**Rating:** 6
**Confidence:** 2

**Summary:**

This paper investigates the limitations of the Contrastive Loss (CL) for training dense neural retrievers and proposes a alternative called Mann-Whitney (MW) loss. The MW loss is a simple yet principled objective that directly maximizes the Area Under the ROC Curve (AUC) by minimizing binary cross-entropy over pairwise score differences. Theoretical guarantees and empirical evaluations demonstrate MW loss’s effectiveness in improving calibration and retrieval performance compared to CL.

**Strengths:**

**Novelty:** The paper addresses a critical issue in dense retriever training and proposes a theoretically sound and empirically effective solution. The focus on global score calibration and AUC maximization is well-justified and has the potential to improve the reliability and applicability of neural retrievers in various domains.

**Evaluation:** The authors conduct extensive experiments on various datasets and models, demonstrating the effectiveness of MW loss in both in-distribution and out-of-distribution scenarios. The results consistently show that MW loss outperforms CL in terms of AUC and standard retrieval metrics.

**Clarity:** The paper is well-written and organized, with clear explanations of the problem, proposed method, and experimental results.

**Weaknesses:**

**Generalization:** Exploring the performance of MW loss on various models would strengthen the paper’s claims and provide a more comprehensive understanding of its generalizability.

**Efficiency:** While the paper mentions the computational differences between MW loss and CL, a more detailed analysis of the computational cost and its impact on training efficiency would be beneficial.

**Other Methods:** While the paper focuses on comparing MW loss with CL, a comparison with other recent approaches for improving retrieval performance, such as margin-based losses or data augmentation techniques, would provide more of MW loss’s advantages or limitations.

**Questions:**

Please see the weaknesses above.

---

> ### Author Response · Authors · 2025-11-26
> **Addressing comments**
>
> We thank the reviewer for their feedback. While we have explored three model sizes for comparing MW and CL, we acknowledge that experimenting with a larger model (based on an LLM) can bring further insight. We also agree that concrete tables/figures on the computational overhead (efficiency) of MW would add valuable insights to our work. We will add these comparisons to our camera-ready version.
>
> In this paper, we set out to show that intra-query comparisons are a strong signal for models to learn metrics that perform well on IR metrics while significantly improving AUC. MW is our proposal, and we have limited our experiments to comparison with InfoNCE to focus on this point. Comparison of MW with other loss functions that could be cast as intra-query comparison is left as future work.

---

### Official Review · Reviewer_Cvfk · 2025-10-23

**Soundness:** 2
**Presentation:** 3
**Contribution:** 2
**Rating:** 4
**Confidence:** 3

**Summary:**

This paper propose MW loss, which directly maximizes AUC by minimizing BCE over score differences between positives and negatives sampled from different queries. They prove MW upper-bounds AoC, show the computation reuses the same embedding/similarity passes as contrastive while doing more pairwise comparisons in the loss, and report consistent gains in AUC with competitive or better MRR/nDCG in-domain and on BEIR. Convergence is slower but argued to reflect the harder objective.

**Strengths:**

1. clear theoretical critique

2 embeddings/similarities are unchanged vs. contrastive; only the aggregation does B(B−1) pairwise comparisons in the loss, which should vectorize well.

3. results show broad AUC gains and frequent MRR/nDCG improvements across MiniLM/XLM-R-Base/Large, with plots of per-dataset gains.

**Weaknesses:**

1. The baseline design largely collapses to CL vs. MW; this isolates the loss but under-probes stronger retrievers or alternative AUC-aware objectives, leaving open whether MW is the best practical route to calibration and top-k retrieval.

2. In-domain improvements on MRR/nDCG for small/base models are modest on average even as AUC rises, so the paper should clarify when AUC gains meaningfully translate to ranking quality.

3. The compute section argues “same embedding/similarity cost,” but the extra B(B−1) BCE terms can still be a real bottleneck; wall-clock, peak memory, and throughput should be reported for typical B, H, corpus sizes. Convergence is slower; this is acknowledged but not quantified against a fixed training budget.

4. Sensitivity to 100/1k, as well as to full-corpus vs. ANN sampling, would strengthen external validity.

**Questions:**

1. Eq. (2) and Lemma 2: clarify the role of temperature τ in ℓ_BCE vs. the τ used in contrastive; report sensitivity curves of AUC/MRR to τ and whether τ→∞ reduces to a hinge-like objective.

2. the AUC protocol picks top-500 corpus negatives; add a sensitivity analysis for 100/500/1000 and a variant using random-plus-hard mixes to show robustness.

3. How stable is the global threshold learned by MW across domains—can a single score cut transfer from NLI-trained models to BEIR subsets without per-task tuning?

4. Do AUC gains predict top-k gains? On which datasets do we see high ΔAUC but flat ΔMRR/nDCG, and why? Any evidence of MW optimizing separation where intra-query ordering is less affected?


5. Any failure modes with multi-relevant queries or table-heavy evidence where global calibration helps less than within-query ranking? Please include targeted analyses.

---

> ### Author Response · Authors · 2025-11-26
> **Addressing questions and clarifying contributions**
>
> Weakness:
>  We thank the reviewer for their feed back. In this work, our goal was to illustrate that learning a global similarity metric is a strong enough signal for models to learn metrics that perform well on IR and have good calibration. Our claim is that CL (which is used almost exclusively for training embedding models) can be safely replaced by MW to obtain better score separation without hurting (and potentially improving) IR metrics.
>
> In our experiments, the extra computation did not incur any additional time per training step. In our training setup (2 GPUs), we found that batch size and GPU communication added a level of noise to the time of each training step such that the extra computation was not detectable.
>
> Questions:
>
> We agree that sensitivity curves for (in MW and CL loss) and the number of top-k corpora used for calculating AUC can be an informative addition, and we commit to adding them in our camera-ready version.
>  2–3. The second table in our paper is designed to show exactly what this question is asking: a retriever trained on three datasets and evaluated out of domain.
>  4–5. This is an interesting line of investigation, which we have not considered yet. In this work, our goal was to illustrate that aiming to learn a global metric allows the model to learn a score that performs on par with CL while substantially improving AUC. Investigating the specific distributions under which MW has an advantage over CL is an interesting direction, which we shall leave for future work.

---

### Official Review · Reviewer_8QJW · 2025-10-26

**Soundness:** 3
**Presentation:** 3
**Contribution:** 1
**Rating:** 2
**Confidence:** 4

**Summary:**

RAG pipelines typically use retrievers based on dual-encoder models. These retrievers are often trained using contrastive learning (e.g., with an InfoNCE loss). However, this type of training suffers from the fact that the resulting relevance scores are not directly comparable across queries — a phenomenon long known in the Information Retrieval community — which makes a simple, globally applied thresholding strategy unlikely to succeed.

To address this, the authors propose the MW loss, based on the Mann–Whitney U statistic (specifically, a differentiable proxy of this statistic). This loss serves as an upper bound on the AoC (Area over the Curve), thereby directly aligning training with a globally calibrated retrieval objective.

**Strengths:**

* The paper is well written, well structured, easy to read and understand.
* The underlying claims are sound, at least within the (limited) context the authors are considering.
* It indeed offers a new loss that ensures consistency of relevance across queries; this implies that complex query-dependent thresholding strategies are no longer needed.

**Weaknesses:**

(1) The main weakness of the paper is that it does not truly optimize what matters (contrary to what its title suggests).

In Information Retrieval (IR) and, in particular, in Retrieval-Augmented Generation (RAG), what ultimately matters is ranking quality, rather than classification accuracy with respect to an artificially defined binary relevance label. In this context, the AUC metric is rarely used to evaluate the quality of a ranking method, for several well-known reasons:

- It does not depend on the actual positions of items in the ranked list, effectively treating the problem as classification rather than ranking.

- It assumes binary relevance, whereas in practice relevance is often graded or continuous.

- It is pairwise rather than listwise, and it is well established that listwise approaches tend to be more effective.

(2) The choice of baselines for comparison is also quite weak: only a single baseline is considered. Since this baseline is not trained using the same loss function, the comparison—at least for the AUC criterion—appears either unfair or trivial.

See the next section (Questions) for suggestions of additional baselines and ablation studies.

**Questions:**

* How do you position your method with respect to the RankNet objective function, which is fairly standard in the (older) Information Retrieval (IR) literature and can be seen as a differentiable proxy of the AUC, albeit computed at the level of a single query?
* Score normalization and calibration methods, typically applied as post-processing steps, are widely used in the IR community to make relevance scores more consistent and comparable across queries. How does your approach relate to, or compare with, these existing techniques?
* Given that the choice of baselines is somewhat limited (only a single method is considered, and arguably not the most representative of current practice in the RAG community), it would have been valuable to include additional ablations or variants, such asvthe use of other differentiable surrogates of the indicator function 1{𝑧≤0} (e.g., hinge loss, etc.);
* the influence of the temperature parameter;
* a deeper analysis of the somewhat counterintuitive observation that the proposed method preserves or even improves within-query (fine-grained) ranking quality. This last point is not obvious, since AoC is not inherently a ranking loss but rather a classification-oriented objective. Ideally, it would have been interesting to explore multi-valued (non-binary) relevance scenarios to validate this claim.

---

> ### Author Response · Authors · 2025-11-26
> **Clarifying contributions and addressing questions**
>
> We thank the reviewer for their thorough feedback. The main contribution of this paper is to illustrate that dual encoders are capable of learning a general metric. Most information retrieval problems can be thought of as metric learning problems, and all recent state-of-the-art retrievers in the RAG community use InfoNCE (contrastive loss) to train dual-encoder retrievers, which corresponds to list comparison (query-based) training.
>
> Our theoretical arguments in this paper show that, for consistent calibration through training, Contrastive Loss is limited while MW is consistent. The experimental contribution of our work was to set up a fair comparison (training a base pretrained model with CL vs. MW) and show that MW is capable of teaching models that perform on par with CL on list comparison metrics (such as nDCG and MRR). This may seem counterintuitive at first, as CL is directly optimizing for list comparison metrics; however, in our opinion this illustrates the power of pretraining in these language models. In this case, MW is aiming to learn a more general metric and posing a harder learning problem, which in turn unleashes the model’s potential, resulting in performance on par with (and slightly better than) CL on list comparison metrics while attaining a much better AUC score.

---

### Official Review · Reviewer_NUjr · 2025-11-01

**Soundness:** 2
**Presentation:** 3
**Contribution:** 3
**Rating:** 4
**Confidence:** 2

**Summary:**

It introduces a Mann–Whitney (MW) loss and proves an upper bound on the Area-over-ROC, so minimizing MW encourages higher AUC

**Strengths:**

It introduces a MW loss, showing that minimizing MW tightens the AoC bound and encourages higher AUC.

**Weaknesses:**

While the paper positions MW loss as a principled AUC-aligned alternative to Contrastive Loss, it does not compare against other well-known pairwise and listwise ranking objectives (e.g., margin ranking loss, triplet loss, RankNet/LambdaRank...).

Some of these losses optimize s(q, p⁺) − s(q, p⁻) directly, which is structurally very close to the proposed MW loss. Without such comparisons, it is difficult to assess whether MW loss provides a fundamentally new advantage.

**Questions:**

While AUC is theoretically meaningful for score calibration, retrieval benchmarks are primarily evaluated using top-k metrics (nDCG, MRR, Recall@k). The gains in these metrics are sometimes small or even mixed. It remains unclear whether optimizing AUC is the right objective for retrieval effectiveness in practice?

---

> ### Author Response · Authors · 2025-11-26
> **Clarifying paper intention and contributions**
>
> Weakness:
>  We thank the reviewer for their feedback. In this paper, our goal was to illustrate a shortcoming of Contrastive Loss, namely that it narrows its scope of comparisons to a single query, which in turn results in poor calibration. We acknowledge that there are other loss functions in deep learning that resemble MW loss; however, to the best of our knowledge, none of these loss functions have been used in the context of retrieval as a means to improve calibration. We took MW as an exemplar of this family of loss functions and agree that comparing MW to other loss functions of the same nature can bring more insight (which we will add in our camera-ready version).
>
> Question:
>  The main experimental contribution of our paper is that while MW loss aims to learn a harder global metric, it still performs on par (or marginally better) than Contrastive Loss on IR-specific metrics (such as nDCG, MRR, and recall), with the additional benefit of having a substantially better ROC.

---

### Meta-Review · Area_Chair_kNqV · 2025-12-09

**Summary:**

This paper introduces a Mann–Whitney (MW) loss and proves an upper bound on the Area-over-ROC, so minimizing MW encourages higher AUC.

### Pros

* This paper has an interesting theoretical perspective.
* The proposed method is simply without requiring complex architectural changes.

### Cons
* Misaligned objective - "What Matters?"
* Weak baselines and limited practical gains
* Efficiency is unacceptable.

### AC's evaluation

1. from reviews and rebuttals

This paper receives 6442, most reviewers agree with rejection. Reviewer 8QJW (2) was the most critical, stating that the paper "does not truly optimize what matters" and that AUC is rarely used to evaluate ranking quality. They strongly criticized the lack of relevant baselines like RankNet. Reviewers NUjr (4) and Cvfk (4) found the comparison against only InfoNCE too limited and noted that the gains in actual retrieval metrics were unconvincing.
Reviewer DCGo (6) gave a marginal pass echoing concerns about generalization and efficiency.
The authors provide very simple and short rebuttals, which is not convincing.

2. from AC's reading

The paper overclaims in its title. While it successfully optimizes AUC, it fails to demonstrate that this leads to better retrieval systems compared to the sota. The disconnect between the optimized metric (AUC) and the evaluation metrics that actually matter to the community (nDCG, MRR) is too large. Crucially, the absence of strong ranking baselines is a fatal flaw. To claim superiority in ranking/retrieval, one must compare against established Learning-to-Rank objectives (like RankNet or LambdaRank), not just contrastive learning. Without this, it is unclear if the benefits come from the specific MW formulation or simply from moving to a pairwise objective. The marginal improvements in downstream metrics do not justify acceptance. Also, the rebuttals are not convncing at all. Luckily, most reviewers share the same points with AC.

**Reviewer Concerns:**

All concerns are outstanding:

1. Misaligned objective - "What Matters?" (8QJW & NUjr): This is the main point of contention. The title claims to optimize "what matters," but reviewers argue that in Information Retrieval (IR), what matters is Top-k ranking quality (nDCG, MRR), not binary classification AUC. Treating retrieval purely as a classification problem ignores the importance of rank positions.

2. Weak baselines (NUjr, 8QJW, Cvfk): The comparison is almost exclusively against InfoNCE (Contrastive Loss). Reviewers pointed out that many established pairwise or listwise ranking losses (e.g., RankNet, LambdaRank, Margin-based losses) share structural similarities with MW loss. Without comparing against these standard ranking objectives, the specific value of MW loss is unproven.

3. Limited practical gains (Cvfk & NUjr): While AUC improves significantly, the gains in standard IR metrics (MRR, nDCG) are often marginal or mixed21212121. This casts doubt on whether optimizing AUC translates to better actual retrieval performance.

4. Efficiency Concerns (Cvfk): The pairwise nature of the loss implies an $O(B^2)$ computational complexity. Reviewers raised concerns about the training overhead, which were not adequately addressed with concrete efficiency benchmarks22.

**Reviewer Scores:**

All reviewers will keep the original scores, since the rebuttals are not convincing at all.

---

### Decision · Program_Chairs · 2026-01-26

Reject